# Diagnostic and Prognostic Potential of Biomarkers CYFRA 21.1, ERCC1, p53, FGFR3 and TATI in Bladder Cancers

**DOI:** 10.3390/ijms21093360

**Published:** 2020-05-09

**Authors:** Milena Matuszczak, Maciej Salagierski

**Affiliations:** Department of Urology, Collegium Medicum, University of Zielona Góra, 65-046 Zielona Góra, Poland; matuszczakmilena@gmail.com

**Keywords:** biomarkers, bladder cancer, tumor markers, prognosis

## Abstract

The high occurrence of bladder cancer and its tendency to recur in combination with a lifelong surveillance make the treatment of superficial bladder cancer one of the most expensive and time-consuming. Moreover, carcinoma in situ often leads to muscle invasion with an unfavorable prognosis. Currently, invasive methods including cystoscopy and cytology remain a gold standard. The aim of this study was to explore urine-based biomarkers to find the one with the best specificity and sensitivity, which would allow optimizing the treatment plan. In this review, we sum up the current knowledge about Cytokeratin fragments (CYFRA 21.1), Excision Repair Cross-Complementation 1 (ERCC1), Tumour Protein p53 (Tp53), Fibroblast Growth Factor Receptor 3 (FGFR3), Tumor-Associated Trypsin Inhibitor (TATI) and their potential applications in clinical practice.

## 1. Introduction: Bladder Cancer Issues and Biomarkers

Bladder cancer is the most common urinary site of malignancy and the second most common reason of cancer deaths from the genitourinary tract after prostate cancer in the United States, with 81,400 new cases and 17,980 deaths in the year 2020 [1]. Globally there are about 430,000 new cases diagnosed each year [2].

Favorably, non-invasive lesions constitute approximately 75–80% of newly diagnosed urothelial bladder cancers (UBC). More than 50% of UBCs are caused by smoking. Other important factors include occupational exposure to aromatic amines and polycyclic hydrocarbons. Less evident is the impact of diet and environmental pollution. Increasing data indicate that genetic predisposition plays a role in UBC pathogenesis [2,3,4].

There are two major groups of patients with distinct prognosis and molecular features.

Carcinoma in situ (CIS) and tumors staged as Ta, T1 are grouped as non-muscle-invasive bladder cancers (NMIBC) [5]. NMIBC patients generally have a significant risk of recurrence and potential clinical course for progression [6] but their life expectancy is long and the cancer rarely progresses to muscle invasion. For NMIBC, the major problem is that after the initial transurethral resection of the bladder (TURB), they characteristically recur in 50–70% of cases, with only approximately 10–20% of cases progressing to muscle-invasive bladder cancer (MIBC) [7].

Muscle-invasive tumors very often metastasize and are usually diagnosed de novo, the prognosis is unfavorable and for decades there has been made no major innovation in therapy. Papillary non-invasive cancers (pTa) grow up from carcinoma in situ (CIS) of the urothelium (frequently TP53-mutated, a high-grade lesion) and often metastasize and evolve into muscle invasion [8]. Robertson et al. demonstrated that MIBC shows high overall mutation rates but fortunately most of them seem to be passenger variation without any functional meaning, or repeated genetic alterations including the *TP53*, *FGFR3*, *PIK3CA* and *RB1* genes’ mutations [4,9].

Muscle-invasive bladder cancer (MIBC) is a high risk but potentially curable disease. Unfortunately, still nearly half of patients die from MIBC despite getting the appropriate treatment [10,11]. The major problem in the management of superficial bladder cancer is its tendency to recur. Lifelong surveillance with a relatively long-life expectancy (5-year survival rate > 90%) makes it the most expensive and time-consuming malignancy to treat.

In recent years, a great effort has been put in the search for new potential biomarkers such as protein 53 (p53), ERCC1, CYFRA 21.1, FGFR3 and TATI in the prognosis and prediction of bladder cancer. The *FGFR3* mutations could be a marker of low-grade and early stage tumors, while the changes in p53 appear better in detecting high-grade or advanced cancers.

## 2. Diagnostic and Prognostic Potential of Bladder Cancer Biomarkers 

### 2.1. Cytokeratin Fragment 21.1 (CYFRA 21.1)

Cytokeratin fragments (CYFRA 21.1) is an ELISA-based assay that detects the concentration of a soluble fragment of cytokeratin 19 by using two monoclonal antibodies [12]. The studies have shown that the differentiation between liquid biopsies of healthy (non-cancer) individuals and BC patients may be done using this biomarker. 

Authors [13] concluded that both serum and urine CYFRA 21.1 present decisive indexes for bladder cancer diagnosis. They made a systematic analysis which indicated the pooled sensitivities and specificities for the serum and urine CYFRA 21.1 were of 42%, 82%, 94% and 80%, respectively. The areas under the receiver operating characteristic curves (AUC) for the serum and urine CYFRA 21.1 were in sequence 0.88 and 0.87 (Table 1). 

In an extensive meta-analysis of three case–control studies Kuang [14] confirmed that urinary or serum samples containing CYFRA21.1 can be used as diagnostic biomarkers and for the distinction between local and metastatic bladder cancer. In this meta-analysis, all healthy individuals had a lower CYFRA21.1 level than patients with bladder cancer. The locally invasive disease showed also lower CYFRA 21.1 levels than the subgroup with metastatic bladder cancer. Notwithstanding, between patients with bladder cancer stage I and stage II, and among the group of patients with local stage II and III were no significant differences in the CYFRA21.1 level. Therefore, CYFRA 21.1 cannot be useful in differentiating grades I–III of local bladder cancer but may be used as a diagnostic biomarker and to detect metastases. 

Nisman [15] evaluated that for detecting transitional cell tumors that were grade 1 with CYFRA 21.1 measured in urine samples gave a three-times higher sensitivity compared with the sensitivity of cytology.

CYFRA 21.1 has a high sensitivity for identifying high-grade and CIS tumors and a greater accuracy for the detection of primary tumors than for the recurrence, but it cannot be used for an early detection of BC. The specificity of this test is between 73% and 86% and the sensitivity is between 70% and 90% [12,15,16].

Andreadis and colleagues [17] analyzed a group of 142 patients with invasive bladder cell carcinoma, including 56 patients with stage T1-4 N0 M0 and 86 with involved lymph nodes or distant metastases. The control group contained 33 healthy volunteers. Seven per cent of patients with the locally advanced disease and 66% of patients with the metastatic disease had an elevated level of this biomarker. CYFRA 21.1 may also be a useful tool in indicating the response to chemotherapy.

Importantly, Nisman and colleagues [15] showed that CYFRA 21.1 detected 100% of CIS, 92.8% of invasive bladder tumors (T2 or higher classification) and 91.9% of grade 3 tumors. The CYFRA 21.1 assay identified almost all tumors (with the exception of only one) that had a positive cytology. Moreover, the assay detected 65% of recurrent tumors and 71% of primary tumors that were omitted by cytopathology.

Unfortunately, CYFRA 21.1 is a false positive in the group of patients with urinary tract infections, stones, history of pelvic radiotherapy, urethral catheterization or BCG intravesical instillation within the three previous months. Even years following intravesical immunotherapy with the BCG level of urinary CYFRA 21.1 may be elevated.

Importantly, the abnormal serum level of CYFRA 21.1 [18] corresponds with a worse response. 

In conclusion, Washino [19] observed that serum CYFRA 21.1 might be a marker of high-grade and advanced urothelial carcinoma. On the contrary, CEA and CA19-9 were not demonstrated as potential tumor markers.

The centrifugation step in the methodology is the very important one to improve the precision of this assay by removing cells’ debris that contains a large amount of CYFRA 21.1, i.e., after this process, a significant decrease in the number of true positive and false positive results can be observed [20].

CYFRA 21.1 is considered as one of the best urinary markers for bladder cancer. Jeong and colleagues noticed that CYFRA 21.1 and NMP22 are the most effective at predicting bladder cancer [21]. However, there is a disadvantage being that the concentrations of both markers are strongly influenced by benign urological diseases, intravesical instillations and also a disappointing performance in low-stage bladder cancer.

**Table 1 ijms-21-03360-t001:** Predictive capacity of bladder cancer biomarkers.

Protein Name	Gene Symbol	Purpose	Diagnostic Value	Prognostic Value	FDA Approved	Method	Samples Used (No. Patients)	Predicitive Capacity	Reference
CYFRA 21.1	*KRT19*	Diagnostic and surveillance	Both serum and urine CYFRA 21.1 levels provide an effective index for the diagnosis of BC.	High risk of malignancy- significantly higher serum level of CYFRA 21.1 according to tumour stage (*p* < 0.01) and grade (*p* < 0.05). Patients with increased CYFRA 21.1 level had significantly worse disease-specific survival (*p* < 0.0001, log rank test) [19]. Moreover, patients with metastases had a higher CYFRA 21.1 level than those with locally invasive BC [14].	No	Meta-analysis performed using STATA 12.0 on the base of studies had published before 2 November 2014 in EMBASE, Web of Science and Medline databases. Quality of the studies was assessed by revised QUADAS tools, all of selected studies were English language publications and evaluate diagnostic accuracy of CYFRA 21.1 in patients with BC. Systematic review included 13 studies and 1,262 BC and 1,233 non-bladder cancer patients. 8 studies measured urine and 5 serum level of CYFRA 21.1. In serum detection of CYFRA 21.1 471 BC and 296 non- bladder cancer patients were analyzed. Urine CYFRA 21.1 studies included 538 BC and 678 non-bladder cancer patients.	Urine (*n* = 538 BC/678 control)	Sensitivity = 82% Specificity = 80% AUC = 0.87	[12,13,14,19]
							Serum (*n* = 471 BC/296 control)	Sensitivity = 42% Specificity = 94% AUC = 0.88	
DNA EXCISION REPAIR PROTEIN ERCC-1	*ERCC1*	Diagnostic and surveillance	71.3% (308/432) of cases was ERCC1 positive. Ta = 3.2% T1 = 11.7% T2 = 21.4% T3 = 45.1% T4 = 18.5% CIS = 8.1% LG = 20.8% HG = 79.2%	ERCC positive tumour had significantly better disease-free survival (HR 0.7, *p* = 0.028) than ERCC1 negative tumours. ERCC1 positive tumours has significantly reduced risk of recurrences (HR 0.71, *p* = 0.021). The 5-year DFS and CSS were better for ERCC1 positive than negative, and were respectively 62% vs 49% and 70% vs 59%. However, there was no important outcomes of adjuvant cisplatin-based chemotherapy by ERCC1 status.	No	Study cohort had 432 patients and 308 of tumours expressed ERCC1. Staining was conducted using Abcam^®^ mouse monoclonal antibody and expression of ERCC1was evaluated by 2 pathologists. Chi-square test was made to assessed differences between ERCC1 expression. All analyses were performed with STATA^®^, version 13.1. Primary tumour samples collected at RC, cells were lysed and total RNA was extracted with Qiagen^®^ kit. ERCC1 mRNA expression was measured by RNA sequencing and confirmed by qPCR using TaqMan^®^ gene expression assays.	UCB cell lines in vitro (*n* = 432)	No data	[22]
TUMOR SUPPRESSOR P53	*TP53* gene	Diagnostic (as a complementary tool) and surveillance	54% (56/103) of cases had TP53 mutations. Ta = 40% T1 = 52% T2 = 80% CIS = 55% LG = 34% HG = 62%	High risk of malignancy-significant difference of TP53 mutations according to tumour stage (*p* = 0.005) and to cellular grade (*p* < 0.001).	No	Sample collection of urine and tumours from 103 patients. Extraction of mRNA was made by Micro mRNA Purification Kit. Then Verso Kit^®^ were used to reverse transcription, amplification was performed by PCR PrimeStar^®^. FASAY assay was used to detect *TP53* mutations in tumour tissues and urinary cells. Statistical test was performed using SPSS software^®^, version 17.	Primary bladder tumours and associated urine (*n* = 103)	Sensitivity = 34% Specificity = 87% PPV = 0.76 NPV = 0.53	[23]
FIBROBLAST GROWTH FACTOR RECEPTOR 3	*FGFR3* gene	Diagnostic (as a complementary tool) and surveillance	36% (37/103) of cases had FGFR3 mutations. Ta = 55% T1 = 29% T2 = 19% CIS = 10% LG = 62% HG = 26%	Low risk of malignancy-negative association of FGFR3 mutations based on tumour stage (*p* = 0.002) and cellular grade (*p* < 0.001) [23]. Low level of FGFR3 expression is an independent predictor of cancer progression and is associated with HG tumours [24].	No *	Sample collection of urine and tumours from 103 patients. Extraction of genomic DNA was performed by QIAamp Viral RNA^®^ Mini kit. Multiplex PCR Kit were used to amplification. Snapshot ^®^ kit was used to detect FGFR3 eight most frequent mutations hotspots in tumour tissues and urinary cells (two independent analysis were carried out). Statistical test was performed using SPSS software^®^, version 17.	Primary bladder tumours and associated urine (*n* = 103)	Sensitivity = 43% Specificity = 98% PPV = 0.94 NPV =0.76	[23]
								Sensitivity = 97.6% Specificity = 84.8% AUC = 0.96 NPV = 0.996 (1)	[25]
TUMOR-ASSOCIATED TRYPSIN INHIBITOR	*SPINK1* gene	Diagnostic and surveillance	49.1% (54/110) of cases had TATI expression. Stage <T2 = 66.7% Stage ≥T2 = 44.9% LG = 76.2% HG = 44.9%	Low risk of malignancy- negative association of TATI expression was positively correlated based on tumour stage (*p* = 0.048) and poor differentiation (*p* = 0.013). Significant differences were observed between TATI-positive and negative specimens in PFS and OS (Log-rank test, *p* = 0.003, 0.003). In a group of patients with BC undergoing RC TATI expression was independent protective factor. Moreover, TATI expression could enhance prognostic value of p53.	No	Study cohort had 110 patients and 54 of tumours, undergone RC, expressed TATI. Staining was conducted using Abcam^®^ anti-TATI monoclonal antibody and expression of TATI was evaluated by 2 pathologists. Proportion of immune-positive cells and their staining intensity was scored in two scales and used to evaluation of TATI expression. All analyses were performed with SPSS software, version 21.	Tissue microarrays from UCB (*n* = 110)	No data	[26]
						Study cohort consisted of 160 patients, divided into 3 groups. Group 1 had 80 primary HG UBC. Group 2 of 40 healthy volunteers and group 3 of 40 benign UBC. TATI was measured using a radioimmunoassay according to the manufacturer’s instructions (Orion Diagnostica). Analyses were performer with STATA^®^, statistical software, version 6.0	Urine (*n* = 160)	Sensitivity = 85.7% Specificity = 77.5%	[27]

(1) Using a logistic regression analysis with a model consisting of the 3 markers’ methylation values, FGFR3 status, age and known smoker status at the diagnosis time. * It is available THERASCREEN^®^ FGFR RGQ RT-PCR KIT. Abbreviations: HR—Hazard Ratio, *n*—number of patients participating in study, *p*—calculated probability, CIS—carcinoma in situ, HG—high grade, LG—low grade, FASAY—Functional Analysis of Separated Allele in Yeast, ELISA—enzyme-linked immunosorbent assay, IHC—immunohistochemistry, RC—radical cystectomy, BC—bladder cancer, CCS—cancer specific survival, DFS—disease-free survival, UCB—urothelial carcinoma of bladder, qPCR—quantitative polymerase chain reaction, PPV—positive predictive value, NPV—negative predictive value.

### 2.2. Excision Repair Cross-Complementation 1 (ERCC1) 

The nucleotide excision repair (NER) pathway is important for the protection of genomic stability and for the removal of platinum-induced DNA adducts and cisplatin resistance [28,29]. The key molecules in this pathway belong to the excision repair cross-complementing group 1 (ERCC1) [30].

The ERCC1 role is detecting, repairing and rate-limiting the interstrand cross-links in DNA [31]. Therefore, this enzyme may be representative for the crucial DNA damage repair ability of the cell [32,33]. In a group of patients treated with a surgical resection, ERCC1 as the DNA repair protein may also be engaged in weakening the malignancy of tumors by reducing the amount of mutations. Moreover, genetic testing of ERCC1 expression levels could personalize the chemotherapy by selecting the patients who would benefit from platinum-based chemotherapy. A variety of tumors, including bladder tumors, show that the ERCC1 level is strongly associated with cisplatin resistance [34].

One of the first reports in the literature presenting the impact of ERCC1 expression on the survival of oncologically treated patients was the study of George R. Simon. In 2005, Simon’s analysis included 51 patients who were operated on for non-small cell lung cancer and determined their ERCC1 expression. The median survival in the ERCC1 positive expression group was found to be significantly longer—94.9 months compared with 35.5 months in the negative ERCC1 group. The conclusions were that the ERCC1 expression might be an independent prognostic factor for survival in lung cancer [32].

In 2006, Olaussen's work on a large group of patients was published, which included the results of a study of 761 patients after radical lung cancer surgery. The goal was to identify a group of patients who might take an advantage from adjuvant treatment. The study showed that the benefit of adjuvant chemotherapy concerned the patients with a negative ERCC1 expression. An interesting finding was that in the group that did not receive chemotherapy but was only treated surgically, patients with a positive ERCC1 expression had a longer survival compared with those with a negative ERCC1 expression [31].

In advanced non-small cell lung cancer, the ERCC1 expression has a significant prognostic value and its high level is associated with a longer survival in patients who do not receive chemotherapy after a complete resection [31,32]. Piljić et al. indicated that the ERCC1 expression in all stages of lung carcinoma has a great value in monitoring patients receiving chemotherapy based on platinum [35]. Li et al. indicated that in a group of patients with advanced non-small cell lung cancer, ERCC1-negative had better progression-free survival (PFS) (*p* = 0.016) and overall survival (OS) (*p* = 0.030) in comparison with positive patients [36].

The value of ERCC1 has also been confirmed in other cancers. ERCC1 is one of the most frequent in 84% or even more of colon cancers, and reductions of a DNA repair gene has been observed [37,38]. In 40% of the crypts within 10 cm on each side of colonic adenocarcinomas, ERCC1 was found to be deficient [37]. The literature data presented above show a significant relationship between the ERCC1 expression and survival in different types of cancer.

In 2012, Sun [39] analyzed 93 patients with BC who underwent radical cystectomy and they demonstrated that ERCC1 can be used as a prognostic and predictive biomarker in this group. An ERCC1-positive expression was found in 58% of patients, and the study group was divided into those who received additional adjuvant chemotherapy and those without chemotherapy. It was found that patients after radical cystectomy without adjuvant chemotherapy with a high ERCC1 expression have a significantly longer five-year survival than those with a low expression, 84% to 49%, respectively. It has also been reported that ERCC1-negative patients potentially may benefit from adjuvant chemotherapy. 

Klatte and colleagues [22] presented the work assessing ERCC1 as a prognostic and predictive biomarker of bladder cancer after cystectomy. In a group of 432 patients, a positive expression was found in 71% of patients. Patients with an ERCC1-positive expression had a significantly better five-year disease-free survival (DFS) than those with an ERCC1-negative expression, 62% to 49%, and cancer-specific survival (CSS), 70% to 59%, respectively. In the ERCC1-positive group, the risk of bladder cancer (BC) recurrence and death due to BC was 30% lower. Patients undergoing radical cystectomy with an ERCC1-positive expression had better survival values than those with a negative expression. Therefore, ERCC1 may be an independent prognostic marker for bladder cancer. 

Similar conclusions were made in Hemdan's report. They evaluated a group of 244 patients who underwent radical cystectomy or neoadjuvant chemotherapy and radical cystectomy. Negative ERCC1 correlated with a worse overall survival in the group with only surgical treatment. It was noted that neoadjuvant chemotherapy would benefit mainly patients with an ERCC1-negative expression, while for those who were ERCC1-positive, the influence was minimal [40].

Another meta-analysis was published by Urun [41], performed on 1425 patients from 13 studies, and patients with an ERCC1-positive expression constituted 24–76% of the examined populations. The role of ERCC1 as a prognostic factor of survival was assessed in patients with advanced bladder cancer treated with platinum-based chemotherapy. The conclusions were that a positive ERCC1 expression is not significantly related to overall survival, but has a significant impact on worse progression-free survival, and may be an indicator of worse survival in patients with advanced bladder cancer, but large prospective studies are needed to consider ERCC1 as a prognostic marker in patients with advanced bladder cancer.

Sakano [42] suggested that, in the group of patients with bladder cancer undergoing a combined trimodality approach, the disease-specific survival might be predicted by the expression of ERCC1 and XRCC1. A positive expression of these molecules was connected with better disease-specific survival rates but further research is needed to confirm these results.

Analyzing the previous studies, gives controversial information about predicting the prognostic role of ERCC1 in the treatment of advanced bladder cancer. In 2018, Eldehna [34] conducted a descriptive study on 80 patients with muscle-invasive bladder cancer (stages T2–T4a) who received platinum-based chemotherapy. The results of their research showed a significant relationship between a platinum-based treatment response and the ERCC1 expression in bladder cancer tissue samples (*p* = 0.013). It was an indicative association between a negative immuno-expression and more favorable outcome but no difference between the ERCC1 expression and mean overall survival or progression-free survival in different immune-expression levels in patients was apparent. Therefore, ERCC1 may be a potential predictive but not prognostic marker and for this reason, genetic testing could personalize chemotherapy by selecting the patients who would benefit from a platinum-based treatment in bladder cancer.

In summary, ERCC1-positive tumors were associated with better prognosis in cases without chemotherapies. However, in cases with chemotherapies, ERCC1-negative tumors were associated with a better outcome. 

The most possible explanation for the above scenario seems related to the function of this enzyme, which appears crucial in the DNA damage repair ability of the cell. The above DNA repair, related to the ERCC1 activity, is, however, non-beneficial for patients treated with chemotherapy, potentially leading to an “antichemotherapeutic” activity.

### 2.3. Tumour Protein p53 (TP53)

The common oncosuppressor gene mutated in all human cancers and the most frequently mutated gene in MIBC is the tumor protein p53 (*TP53*) [43]. Genomic integrity and stability are maintained by *TP53* via triggering a cell-cycle arrest, apoptosis, autophagy and DNA repair. Mutant p53 proteins silence the autophagy related gene (ATG) which affects the autophagic flow, and therefore suppresses regulation to the autophagic vesicles formation and their fusion with lysosomes [44]. Additionally, p53 preferentially binds to the AMPKα subunit and inhibits the AMPK activation. Mutp53s become oncogenic via the activation of AMPK [45]. 

Bladder carcinogenesis is closely associated with tumor suppressor dysfunction and the inactivation of *TP53* [46]. Therefore, p53 has been studied as a marker of urothelial cell carcinoma recurrence and progression. 

Cheap and simple methods to detect the abnormal function of p53 is immunohistochemistry staining (IHC). The short half-life of wild-type p53 prevents its intra-nuclear accumulation [47].

Increased p53 accumulation in the cell nucleus is a result of *TP53* mutations. 

Immunohistochemical patterns of *TP53* mutations are strongly associated with the progression of urothelial cell carcinoma. Plenty of data illustrate that from non-missense mutations (i.e., nonsense, insertion and deletion) to wild-type *TP53*, the expression of p53’s IHC increases. That promotes the grow up of an invasive phenotype of bladder cancer [48]. The high expression of p53 has been associated with features of tumor aggressiveness and correlated with poor oncological outcomes [43,49]. Therefore, this protein level was higher in more advanced bladder cancer [50,51]. Plenty of studies have indicated that p53 can be useful to assess the level of progress and to prognose urothelial cell carcinoma [49,51].

However, Ciccasese and colleagues [52] published a study with a contradictory opinion. In their opinion, the single p53 marker is not good enough as a prognostic marker of MIBC. 

Moreover, the most aggressive T1 high-grade cancers appear to be also associated with the expression of this protein. The progression from T1 NMIBC to T1HG can be predicted by a p53 overexpression [53]. 

Authors [51] collected data from 70 patients and showed that 16% of patients with low-grade and 91% of patients with high-grade lesions were p53-positive. There was 33% positivity in Tis, 55% in T1, 72% in T2 and 100% in T3a and T3b. These results indicated a strong intensification of p53 staining—94.6% of high-grade and 5.4% of low-grade tumors. Moreover, the p53 accumulation in the nucleus, in a group treated with radical cystectomy and in other MIBCs, has a prognostic value [54].

Another study showed that an aggressive tumor phenotype is strongly associated with the overexpression of p53 [43]. MIBC and CIS correlated with a high level of *TP53* deletion and mutation [55]. According to the TCGA cohort data [4], 89% of MIBCs have an inactivated TP53 cell-cycle pathway, with *TP53* mutations in 48%. Bladder epithelial cells become malignant by the TP53/RB1 pathway or the FGFR3/RAS pathway [55]. 

### 2.4. Fibroblast Growth Factor Receptor 3 (FGFR3)

Fibroblast growth factor receptor 3 (FGFR3) alternations are associated with urothelial cell carcinoma pathogenesis [56,57]. FGFR3 is activated by the mutation or overexpression in many bladder tumors at any stage, but is predominantly active in low-grade NMIBCs [58,59]. Higher levels of FGFR3 expression were observed in low-grade, non-invasive tumors and recurrent non-invasive tumors than in invasive and non-invasive high-grade carcinoma [57].

This marker is associated with a lower chance of progression to a muscle-invasive disease and it is like a hallmark of the low-grade pathway. FGFR3 alternations occur mainly in non-invasive tumors [59,60], specifically in the luminal-papillary subtype (35%), which has the best overall survival and is characterized by a papillary morphology [59,61]. Moreover, many studies indicated that *FGFR3* mutation and the risk of progression are an inverse interaction. Therefore, patients with MIBC and the *FGFR3* mutation have better survival rates [62]. Another study suggests that also the progression in pT1 tumors is in negative correlation with the *FGFR3* mutation [63]. Many studies confirm that *FGFR3* mutations correlate with an overall benign effect [59,63,64]. Moreover, in the risk stratification, surveillance and diagnosis of low- or high-risk NMIBC patients, *FGFR3* mutations combined with the promoter hyper-methylation of *HS3ST2*, *SEPTIN9* and *SLIT2* have shown 97.6% sensitivity and 84.8% specificity (Table 1) [25]. The presence of the *FGFR3* mutation in urine is observed not only in low-grade tumors but it also seems to be associated with future recurrence [65,66]. 

FGFR3 is involved in tumorigenesis in ∼40% of invasive bladder cancer and in the majority (∼80%) of low-grade non-invasive (stage Ta) bladder cancers [59]. Tomlinson et al. observed an FGFR3 overexpression in nearly 40% of MIBC, whereas mutations occurred in 21% of MIBC [67]. Sung [56] observed that an FGFR3 overexpression results in the worst overall survival and disease-free survival in a group of patients with adjuvant chemotherapy. In a group without this treatment, no prognostic significance was observed.

High levels of FGFR3- and PIK3CA-mutated DNA in urine can be useful in predicting later metastasis and progression in NMIBC [68]. Choi et al. indicated that *FGFR3* mutations are characteristic for the luminal type of MIBC [69]. In conclusion, FGFR3 may be an important therapeutic target in both non-invasive and invasive BC [58,59].

In the results of their research, Beukers [60] confirmed that mutations in *FGFR3* were more often observed in low-grade tumors and the papillary urothelial neoplasm of low malignant potential (PUNLMP) + G1 (61.9%) than in high-grade tumors G2 + G3, at 17.2%. It was also observed that *FGFR3* mutations were more frequent in non-invasive tumors’ Tis and Ta stages, at 53.4%, than in the invasive stages of T1 and T2, at 12.5%. Mutations correlated with a better survival rate and occurred in a higher level in non-invasive than in advanced diseases, and these values for TaG1, TaG2, TaG3 + T1 and T2 were 67.3%, 43.3%, 20.3% and 6.3% respectively. It was also noticed, but with no statistically significant correlation, that *FGFR3* mutations increase the possibility of disease recurrence [70]. Hosen et al. showed that *FGFR3* mutations have no significant influence on patient survival and that in the Ta, T1, TaG1 and TaG2 diseases, it did not significantly predict the recurrence rate [70]. 

Knowles et al. also demonstrated that the *FGFR3/HRAS* mutation was often present in the development of urothelial hyperplasia, which can progress to non-invasive papillary tumors with high recurrence rates via the FGFR3/RAS pathway [8].

Van Rhijn [63] conducted a study on a group of 132 patients with primary pT1 bladder cancer. The diagnosis was confirmed after a uropathologist review of the slides. *FGFR3* mutations were identified by a SNaPshot^®^ analysis in 37 of 132 pT1 bladder cancer cases (28%) and an altered P53 expression was determined by standard immunohistochemistry in 71 of them (54%). Both molecular alternations were observed in 8% of patients. In predicting progression, carcinoma in situ and the status of the *FGFR3* mutation were significant but *TP53* was not. It was also mentioned that the presence of *FGFR3* mutations helps to identify patients who have a better disease prognosis because the *FGFR3* mutation occurs with lower grade and altered *TP53* with high-grade pT1 bladder cancer. 

Hernández and colleagues [64] analyzed 772 samples from patients with bladder tumors reviewed by expert pathologists. Their results indicated that *FGFR3* mutations were more frequently observed in neoplasms with low malignant potential, at 77%, and in tumors TaG1, at 61%, and TaG2, at 58%, than in tumors TaG3, at 34%, and T1G3, at 17%. They also confirm the association between superficial tumors and a high presence of recurrence. Nevertheless, a significant increase risk was observed only in the group of patients with TaG1 tumors. In this study, another positive correlation of good prognosis and occupancy of FGFR3 was confirmed.

Kompier [71] performed a study on 118 patients with primary and recurrent NMI-BC. They analyzed the *FGFR3* mutation status in the disease process. The analyzed group had 2133 cystoscopies done within the median follow-up of 8.8 years and 414 tumor recurrences developed in 80 patients. *FGFR3* mutations were equally distributed in the recurrences and the primary tumors (63%). Different tumors may have a variety of *FGFR3* mutations types. Mutant or wild-type primary tumors had a similar risk of recurrence but in 81% of recurrences, a mutation was found. In this group, recurrences developed after 10 years and, in comparison with the wild-type primary tumor, occurred in a lower grade and stage.

Therefore, a follow-up surveillance based on the presence of the *FGFR3* mutation analysis with the reduction in the number of cystoscopies may be considered [71].

In another study, Kompier and colleagues [72] confirmed the correlations between a low risk of progression and better disease-specific survival in the primary mutant FGFR3 tumor and worse prognosis in the group of patients with an overexpression of p53.

Williams et al. found that in the selection of patients for the FGFR-targeted therapy, the existence of a fusion protein, which indicates other classes of mutations in a group with a high FGFR3 expression, may be helpful [58].

The study [73] shows that *FGFR3* mutations may influence tumorigenesis by regulating an acute inflammatory response which via the immune cells destroys the tumor cells. Therefore, there may be potential treatment strategy for the early stage of FGFR3-mutated or overexpressed BC based on the synchronal inhibition of FGFR3 and the immune modulators.

Noel [23] conducted a pilot study to assess the *TP53* and *FGFR3* mutations in urine and tumoral tissues samples that had been collected from 103 BC patients. Mutations in *TP53* were detected in 54% of the 103 bladder tumors and the distribution increased with the cellular grade (*p* < 0.001). The *TP53* mutation presented 34% of low- grade (LG) and 62% of high- grade (HG) tumors. The potential prognostic value of *TP53* may indicate a significant difference in the tumor stage (*p* = 0.005). The specificity was 87%, with the positive predictive value (PPV) 76% and with the negative predictive value (NPV) 53%. However, the sensitivity in the urine test was only 34% (Table 1).

In 36% of analyzed tumors, *FRFG3* mutations were identified and their distribution decreased with the cellular grade (*p* < 0.001). They occurred in 62% of LG tumors versus 26% in HG. A negative correlation was also between the *FGFR3* mutations and tumor stage (*p* = 0.002). All predictive capacities were better for the *FGFR3* than for the *TP53* mutations measured in this study, the sensitivity was 43% and the specificity was 98%, with the PPV 94% and the NPV 76% (Table 1) [23].

The results showed that TP53/FGFR3 could be useful as a complementary tool in diagnosis but could not replace urine cytology. The tumor stage and grade are strongly correlated with the *FGFR3* and *TP53* mutations, which are in “mirror distribution” [23].

Kang [24] enrolled 120 patients with primary pT1 BC and examined in this subgroup the utility of expression levels and mutation status of *FGFR3* as a prognostic marker. In this study, 40% of patients had *FGFR3* mutations and those patients also had significantly higher levels of the FGFR3 expression compared with the FGFR3 wild-type BC (*p* < 0.001). The mutation status was not associated with cancer progression, but a low level of FGFR3 correlated with cancer progression and HG tumors (*p* = 0.001 and *p* = 0.006). Therefore, the FGFR3 expression level was, in the multivariate analysis, identified as an independent predictor of cancer progression (Table 1). Significant was also the correlation between the *FGFR3* mutation and a low tumor grade. In tumor recurrence, both the *FGFR3* mutation status and mRNA expression level revealed no significant differences (*p* = 0.264 and *p* = 0.856, respectively).

In conclusion, FGFR3 may be used as a urine-based assay in the detection of primary tumors, recurrences, for prognosis and targeted therapies. 

### 2.5. Tumor-Associated Trypsin Inhibitor (TATI)

TATI is a peptide produced at lower concentrations in many healthy tissues, especially in the gastrointestinal and urogenital tracts but also in the gall bladder, kidney and breast. 

It occurs in high concentrations by several tumors such as gynecologic, gastrointestinal, urologic, lung, breast, head and neck cancers [74,75,76,77,78,79]. An increased level of TATI is also observed in renal failure and in dialysis patients because this peptide is cleared from the circulation by renal excretion. Therefore, a low glomerular filtration rate correlates with an increase in TATI [80].

TATI is connected with tumor aggression because it appears in the co-expression with tumor-associated trypsin, which participates in moderating tumor-associated protease cascades [81]. 

TATI is produced at high concentrations by mucinous ovarian tumors, and was initially isolated from the urine of a patient with ovarian cancer. The most useful clinical application of this peptide is observed in the detection of ovarian tumors: benign and malignant [82]. 

TATI occurs in a high level also in other benign and malignant diseases. Pancreatitis and strong acute phase reactions (when serum CRP is clearly increased (>90 mg/L)) such as severe injury or inflammatory diseases trigger a TATI expression. This fact is a limiting factor of the use of TATI as a tumor marker but it does not invalidate this peptide [83]. In cancers, an increased TATI concentration is associated not only with tumor production but also acute phase reactions caused by tissue destruction during cancer invasion [81].

Serum values of TATI have also been used in patients with muscle-invasive and metastatic transitional cell carcinoma, to monitor the response to therapy. In 1996, Pectasides [74] suggested that TATI might be potentially useful in monitoring the efficacy of treatment in transitional cell carcinoma of the bladder. Significantly modified values of TATI were observed in metastatic diseases, in patients with complete or partial remission and non-responders. An important increase in TATI in T2-T4-N0M0 tumors were in the non-responders.

Kelloniemi et al. showed that for the identification group of patients with adverse prognosis in transitional cell carcinoma serum, TATI might be an independent prognostic factor [84].

Shariat [85] indicated that TATI is more specific than NMP22 for the detection of bladder transitional cell carcinoma (TCC). They showed also that higher levels of TATI were in TCC patients and in more invasive stages.

In 2006, Hotakainen [86] reported that a TATI expression was observed in all non-invasive tumors and benign tissues, but the expression was lower in the muscle-invasive tumors. Therefore, they concluded that the TATI expression decreases with the rising stage and grade of the tumor in bladder cancer. Therefore, as for TATI, Shariat [85] showed that higher levels of TATI were associated with more invasive TCC but Hotakainen [86] revealed that the TATI expression decreases with the rising stage. The discrepancy between the results of the studies is most probably related to the different populations of bladder cancer patients. The study by Shariat [85], comprised of 153 consecutive patients who had a history of previous, histologically confirmed bladder cancer, without evidence of muscle invasion (stages Ta, T1 and/or CIS). In the Hotakainen (*n* = 28) group, the individuals were affected with both non-invasive and invasive BC.

Gkialas [27] showed that TATI was significantly more sensitive in stage Ta (80%) than was CYFRA 21-1 (32%), UBC (12%) and cytology (20%). TATI was different also between stages and was more sensitive compared with other tumor markers for stage T1.

Patschan and colleagues [87] confirmed that the TATI level shows a positive correlation with low-stage tumors and the favorable differentiation of bladder cancer. They also showed in univariate analyses, that a decreased level of TATI was associated with high recurrences and cancer-specific mortality.

Liu [26] made a similar conclusion that a decrease in the TATI expression correlated with a more advanced disease. Moreover, in the progression of bladder cancer, the prognostic value of a p53 overexpression can be enhanced by TATI.

Bladder cancer management is one of the most complex and expensive in uro-oncology. An ideal biomarker of the future should be potentially able to detect the disease before its clinical manifestation. The BC mortality rate is another major reason to obtain a similar screening method to that available in other cancers, i.e., prostate and colon. 

Currently, flexible cystoscopy remains a mainstay in BC diagnosis and it appears unlikely that available biomarkers would quickly rule out this standard approach in clinical practice. On the other hand, developing markers showing a correlation with cancer aggressiveness and being able to distinguish between aggressive and non-aggressive tumors appears of utmost clinical importance. Hopefully, one of the discussed markers might become helpful in patients’ selection for an appropriate treatment plan and personalized cancer medicine. The prospective studies on a larger group of individuals are still needed in order to obtain additional prognostic information that will improve results, reduce adverse effects and in future allow us to individualize bladder cancer treatments.

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
