# Peer review of "Diagnostic and Prognostic Potential of Biomarkers CYFRA 21.1, ERCC1, p53, FGFR3 and TATI in Bladder Cancers"

_ijms, 2020, doi:10.3390/ijms21093360_

Round 1

Reviewer 1 Report

This review just pasted previous reports. Authors should summarize the reports with their interpretation. Overall, the structure of the text is poor, and it is difficult to read and understand what authors would like to state. The text should be organized.

As for TATI, Shariat et al showed that higher levels of TATI were associated with more invasive TCC. But, Hotakainen et al revealed that TATI expression decreases with rising stage. Why were the opposite results observed? Authors should explain the mechanism clearly. Otherwise, the descriptions were so confusing.

As for ERCC1, ERCC1 positive tumors were associated with better prognosis in cases without chemotherapies. But, in cases with chemotherapies, ERCC1 negative tumor were associated with better outcome. Please clearly state the mechanism of the difference.

English should be modified. There are a number of incorrect grammars and expressions. Comma should be added in many parts of the text.

Author Response

Dear Reviewer, 

Thank you for your input. We did our best to incorporate the suggested changes. 

We summarised our results and we added the paragraphs explaining the discrepancy between the studies on ERCC1 and TATI. 

ERCC1 positive tumours were associated with better prognosis in cases without chemotherapies. But, in cases with chemotherapies, ERCC1 negative tumours were associated with better outcome.

The most possible explanation for the above scenario seems related to the function of this enzyme, which appears crucial in DNA damage-repair ability of the cell. The above DNA repair, related to the ERCC1 activity, is however non-benefial for patients treated with chemotherapy potentially leading to ‘antichemotherapeutic’ activity.

Shariat et al. indicated that TATI is more specific than NMP22 for detection of bladder transitional cell carcinoma (TCC). They showed also that higher levels of TATI were in TCC patients and in more invasive stage [82].

In 2006 Hotakainen et al. reported that TATI expression was observed in all non-invasive tumours and benign tissues, but the expression was lower in the muscle-invasive tumors. Therefore, they concluded that TATI expression decreases with rising stage and grade of the tumor in bladder cancer [83]. Therefore, as for TATI, Shariat et al showed that higher levels of TATI were associated with more invasive TCC but Hotakainen et al revealed that TATI expression decreases with a rising stage. The discrepancy between the results of the studies is most probably related to the different population of bladder cancer patients. The study by Shariat et al., comprised 153 consecutive patients who had a history of previous, histologically confirmed bladder cancer, without evidence of muscle invasion (stages Ta, T1 and/or CIS). In the Hotakainen (n=28) group the individuals were affected with both non-invasive and invasive BC.

Table 1 was changed accordingly.

English editing was performed.

Reviewer 2 Report

The paper cover enough material to be of some clinical use, but the grammar and organization are very poor making it difficult to read and comprehend without going over section repeatedly.

Author Response

Dear Reviewer, 

Thank you for your input. We did our best to incorporate the suggested changes. 

We summarised our results and we added the paragraphs explaining genomic mechanisms in bladder cancer.

English editing was performed.

Table 1 was changed accordingly.

Reviewer 3 Report

This paper is a well-organized review article of vast contents.

There are a few things that need to be corrected.

Table 1 is not well represented of all contents. Diagnosis, prognosis, samples, experimental methods, etc. should be re-created by subdividing them.

Reference 87-92, which is not seen in the text, is cited only in the table, moreover, it does not appear to be related to the content.

The following paper should be cited in the FGFR3 part. {Oncol Lett. 2017 Sep;14(3):3817-3824.}

Briefly explain the Gene function of each gene and the mechanism in bladder cancer.

Superficial bladder cooler, NMIBC is mixed. Unification of terminology should be needed.

Author Response

Dear Reviewer 3, 

Thank you for your input. We did our best to incorporate the suggested changes. 

We added the suggested paragraph /please see below/ and we cited the suggest paper  /Oncology Letters/

Noel et al. conducted a pilot study to assess TP53 and FGFR3 mutation in urine and tumoral tissues samples that had collected from 103 BC patients. Mutations in TP53 were detected in 54% of the 103 bladder tumours and distribution increased with cellular grade (p<0.001). TP53 mutation presented 34% of LG and 62% of HG tumours. Potential prognostic value of TP53 may indicate a significant difference in tumour stage (p=0.005). The specificity was 87%, with the PPV 76% and the NPV 53%. However, the sensitivity in urine test was only 34% (Table 1.) [87].

In 36% of analyzed tumours FRFG3 mutations were identified and they distribution decrease with cellular grade (p<0.001). They occurred 62% LG tumours versus 26% in HG. Negative correlation was also between FGFR3 mutations and tumour stage (p=0.002). All predictive capacities were better for FGFR3 than for TP53 mutations measured in this study, the sensitivity was 43% and the specificity was 98% with the PPV 94% and the NPV 76% (Table 1.) [87].

Results showed that TP53/FGFR3 could be useful as complementary tool in diagnosis but could not replace urine cytology. Tumour stage and grade are strongly correlated with FGFR3 and TP53 mutations which are in “mirror distribution” [87].

Kang et al. enrolled 120 patients with primary pT1 BC and examined in this subgroup the utility of expression levels and mutation status of FGFR3 as a prognostic marker. 40% of patients had FGFR3 mutations and those had significantly higher level of FGFR3 expression compared with FGFR3 wild-type BC (p<0.001). Mutation status was not associated with cancer progression but low level of FGFR3 correlated with cancer progression and HG tumours (p=0.001 and p=0.006). Therefore, FGFR3 expression level was in multivariate analysis identified as an independent predictor of cancer progression (Table 1.). Significant was also correlation between FGFR3 mutation and low tumour grade. In tumor recurrence both FGFR3 mutation status and mRNA expression level revealed no significant differences (p= 0.264 and p=0.856) [88].

In conclusion FGFR3 may be used as urine- based assay in detection of primary tumours, recurrences, for prognosis and targeted therapies. 

We summarised our results and we added the paragraphs explaining genomic mechanisms in bladder cancer.

English editing was performed.

Table 1 was changed as suggested.

Round 2

Reviewer 1 Report

I don't have any more comments.

Author Response

Dear Reviewer, 

Thank you for your input.

Kind regards, 

MS

Reviewer 2 Report

The paper is much improved.  There are still a few errors.  Line 92-Importantnly; line 110-ofthis; line 115-colleugues;

line 239-tumor genesis; line 335-analized.  These are just a few I caught going through quickly.  Please, make one more check.

The paper could really use a concluding paragraph to wrap up the content.

Author Response

Dear Reviewer, 

Thank you for your valuable comments and input.

We tried to correct the misspellings and we did add some concluding remarks.

Kind regards,